# Molecular Dynamics Modeling of Pulsed Laser Fragmentation of Solid and Porous Si Nanoparticles in Liquid Media

**DOI:** 10.3390/ijms241914461

**Published:** 2023-09-23

**Authors:** Irina A. Kutlubulatova, Maria S. Grigoryeva, Veronika A. Dimitreva, Stanislav Yu. Lukashenko, Andrey P. Kanavin, Viktor Yu. Timoshenko, Dmitry S. Ivanov

**Affiliations:** 1P. N. Lebedev Physical Institute of Russian Academy of Sciences, Leninskiy Prospekt, 53, 119991 Moscow, Russia; i.kutlubulatova@lebedev.ru (I.A.K.); grigorevams@lebedev.ru (M.S.G.); lukashenko13@mail.ru (S.Y.L.); kanavinap@lebedev.ru (A.P.K.); vtimoshe@gmail.com (V.Y.T.); 2Institute of Engineering Physics for Biomedicine (PhysBio), Moscow Engineering Physics Institute (MEPhI), 115409 Moscow, Russia; veronikadimitreva@yandex.ru; 3Institute for Analytical Instrumentation of the Russian Academy of Sciences, Rizhsky Prospekt, 26, 190103 St. Petersburg, Russia; 4Department of Solid State Physics, Lomonosov Moscow State University, Leninskie Gory, 119991 Moscow, Russia

**Keywords:** molecular dynamics, modeling, ultrashort laser pulses, fragmentation, nanoparticles, ablation in liquids

## Abstract

The production of non-toxic and homogeneous colloidal solutions of nanoparticles (NPs) for biomedical applications is of extreme importance nowadays. Among the various methods for generation of NPs, pulsed laser ablation in liquids (PLAL) has proven itself as a powerful and efficient tool in biomedical fields, allowing chemically pure silicon nanoparticles to be obtained. For example, laser-synthesized silicon nanoparticles (Si NPs) are widely used as contrast agents for bio visualization, as effective sensitizers of radiofrequency hyperthermia for cancer theranostics, in photodynamic therapy, as carriers of therapeutic radionuclides in nuclear nanomedicine, etc. Due to a number of complex and interrelated processes involved in the laser ablation phenomenon, however, the final characteristics of the resulting particles are difficult to control, and the obtained colloidal solutions frequently have broad and multimodal size distribution. Therefore, the subsequent fragmentation of the obtained NPs in the colloidal solutions due to pulsed laser irradiation can be utilized. The resulting NPs’ characteristics, however, depend on the parameters of laser irradiation as well as on the irradiated material and surrounding media properties. Thus, reliable knowledge of the mechanism of NP fragmentation is necessary for generation of a colloidal solution with NPs of predesigned properties. To investigate the mechanism of a laser-assisted NP fragmentation process, in this work, we perform a large-scale molecular dynamics (MD) modeling of FS laser interaction with colloidal solution of Si NPs. The obtained NPs are then characterized by their shape and morphological properties. The corresponding conclusion about the relative input of the properties of different laser-induced processes and materials to the mechanism of NP generation is drawn.

## 1. Introduction

The role of nanoparticles (NPs) in biomedical application has dramatically grown over the last 20 years [1,2]. Especially widespread the use of NPs was found in the diagnosis and treatment of cancerous tumors [3,4]. The applications of NPs in biomedicine, for example in photoimaging, biosensing, and targeted delivery [5,6,7,8], however, frequently require their parameters such as shape, mean size, magnetic and optical properties to be quite specific, and their colloidal solution must be of high purity (non-toxic). Of high interest for bio-medical applications, in particular, Si NPs have recently made dramatic progress in production due to their biocompatibility [9] and their efficient surface chemistry modification [10]. In work by Zabotnov et al. [11] it was shown the effective fluorescence and light scattering of laser-ablated Si nanoparticles in the visible and near infrared ranges, which opens up new prospects for their use as contrast agents in biophotonics. Compared to other classes of nanoparticles that have significant toxicity, Si nanoproducts, due to their biocompatibility, biodegradability, and specific optical properties can be used for bioimaging [5,6]. The high contrast of local optical properties provided by the accumulation of silicon nanoparticles makes it possible to noninvasively control their biodistribution using optical imaging methods. Light activation of nanostructures accumulated in tumors provides an effective non-invasive treatment method with minimal impact on surrounding normal tissues [3]. Containing an oxide layer, Si-based NPs readily undergo conjugation with various molecules and may play a direct or supporting role in the development of nanomedicines intended to destroy bacteria or viruses [12]. Also, Si-based NPs can expose a high porosity that makes their large surface area highly efficient for the bonding of drugs for their subsequent delivery [13,14].

There are various methods of obtaining nanoparticles: mechanical [15], two-stage synthesis [16], electrochemical, sonochemical, thermal, and photochemical reduction [17,18,19,20], etc. Most of these methods, however, are still at the development stage, since they do not allow obtaining a stable and repeatable result due to problems with aggregation, morphology, growth control, and nanoparticle size distribution. In addition, an important issue is the extraction and purification of synthesized NPs with a focus on their further use [21].

One of the leading methods in the efficient production of environmentally friendly NPs has become pulse laser ablation in liquids (PLAL), which has shown impressive progress over the past 15 years [22] due to its versatility and relative cost-effectiveness in the synthesis of various nanomaterials [22,23]. This method is based on a natural production of nanoclusters during the interaction of powerful pulsed laser radiation with a solid target of virtually any material [24,25], which can then be released to a gaseous or liquid medium to form a nanostructured thin film [26] or a colloidal nanoparticle (NP) solution [27], respectively. Since PLAL procedure does not require the use of specific chemical products and conditions, they are free of limitations of colloidal chemistry and can provide nanomaterials of controllable geometry and composition, which can be used in a variety of important applications, including biomedicine [7,8,28,29]. Another advantage of the laser ablation technique, in comparison with the above methods for obtaining nanoparticles, is the absence of chemical reagents in solutions. In other words, this method makes it possible to obtain chemically pure colloidal solutions of NPs, free from extraneous impurities and radicals [30]. Due to their significantly higher purity compared to chemically synthesized counterparts and their photoluminescent properties, laser ablation and fragmentation nanoparticles can be used for in-vivo biological applications such as vector drug delivery [31,32], imaging [31,33], and therapy [34,35,36,37].

Due to a number of complex and interrelated laser-induced processes during PLAL, however, the obtained NP solution can have broad size distribution, including large (~μm scale) droplets/layers with irregular geometry [38,39]. To reduce the size of NPs obtained by ablation in a liquid, laser fragmentation technologies are used: colloidal solutions of nanoparticles are subjected to additional processing with long (nanosecond) or short (pico- and femtosecond) laser pulses. In this case, as shown in a number of experiments, both nanosecond [40] and femtosecond [41] pulses can be used to affect the final NPs sizes and size distribution, suitable for biomedical needs. In work by Blandin et al. [42], the possibility of fast synthesis of non-aggregated, low-dispersion crystalline silicon NPs was demonstrated by fragmentation under the action of femtosecond laser pulses of preliminarily prepared silicon microcolloids, by means of changing the initial concentration of which, one can control the size and oxidation of the NPs surface.

The performance of the laser-assisted fragmentation method, however, and the characteristics of the final product depend on many parameters, including the laser wavelength, pulse duration, energy density at the target surface, irradiation time, and the composition of the working fluid. Despite some experimental works on fragmentation of silicon NPs, practically little is known about the mechanisms responsible for the formation of final NPs during the fragmentation process. As possible mechanisms, the following are considered: the decay of a molten metal drop obtained by heating with laser radiation due to hydrodynamic instabilities arising due to the asymmetry of the vapor cloud of the surrounding liquid [43], fragmentation of a multiple-charged nanoparticle as a result of a Coulomb explosion [44,45,46], evaporation of atoms when nanoparticles are heated by laser radiation to a temperature above the critical point [47], and fragmentation of nanoparticles under the action of femtosecond laser pulses due to Rayleigh instability [48]. The above models do not take into account such properties as porosity and morphology of nanocrystals, and the laser-induced phase transition processes rely on a number of assumptions and approximations. Some advanced models target accurate descriptions of the short laser pulse interaction with semiconductor solids on the example of Si [49,50] due to several numerical obstacles, where it is limited to very small systems or/and to 1D diffusion models. Thus, it is highly demanded from the point of their technological use to study the physical mechanisms of the interaction of laser pulses with crystalline and porous NPs of Si on a microscopic scale, accounting for the crystal structure and explicitly considering the morphology of the original NPs. In order to gain the knowledge of the optimal laser modes (intensity and duration of laser pulses) for the controlled fragmentation of the initial NPs to the final NPs with the required size and properties, it is also necessary to research the dependances of the final colloidal solution properties on the parameters of the laser pulse as well.

In this work, therefore, we perform large-scale parallel MD simulations for investigation of the mechanism of colloidal NP fragmentation subject to possible manipulation with the final product characteristics via laser irradiation parameters. Because of the explicit atomistic representation due to MD, both the NPs and the liquid media can be characterized with macroscopic thermodynamic properties (temperature, pressure, and density) at any moment of time in the assumption of their local equilibrium within a finite volume ~ 1 nm^3^. Careful analysis of the microscopic properties dynamics allows for the extraction of the mechanism of the NP fragmentation and the possibility of setting up the experiment for NP generation with predesigned properties.

## 2. Results and Discussion

In the scratch the fragmentation process scenario can be decomposed to the following steps: (a) The absorption of a femtosecond laser pulse leads to an increase of free carriers’ density (due to one- or multi-photon absorption process) and the temperature of the electron-hole pairs subsystem goes up to tens of thousands degrees Kelvin, while the lattice temperature remains unchanged; (b) The energy of hot carriers is then passed to the lattice vibrations (phonons) due to the electron-phonon energy exchange, which normally varies in the range of 1–3 ps for Si (the characteristic time of the electron-phonon energy equilibration in Si [51]). In our approximation this time is assumed 1 ps, and the lattice heating due to the electron-phonon interaction is modelled by homogeneous release of the absorbed energy over the substance of the particle from 1st to 3rd ps. The energy absorbed in our approximation (see the Model description) is equal to the incident fluence J/cm^2^ multiplied by the cross section of the particle; (c) As a result of the electron-phonon energy exchange the electron subsystem is cooled, and the lattice is heated, and the irradiated NP can undergo a phase transition (melting, vaporization, and ablation). The heating of the liquid layers surrounding the ablating (fragmenting) NP due to the processes of ordinary phonon heat conduction leads to the formation of a vapor with a transfer of large amount of thermal energy to the heat of vaporization. The surrounding liquid also apply a significant mechanical resistance to the expanding cavitation bubble [38,39,52]; (d) The subsequent formation of new particles is governed by the process of water particles penetrating the expanding bubble, Si vapor nucleation and aggregation of Si clusters. The latter can also lead to the process of voids capturing and formation of new pores inside the new NPs of Si (this assumes negligible effect of surface tension of liquid Si if the new NPs are large enough). The process of emission of the electrons from NPs and related to this accumulation of the positive charge with subsequent Coulomb explosion of the NPs [38,39,53] are not taken into account in our modeling.

The first simulation of fs laser pulse interaction at the incident fluence of 0.015 J/cm^2^ with 3 monolithic NPs of different diameters: 10 nm, 20 nm, and 30 nm in aimed on the identification of a general mechanism of fragmentation process around the threshold value of the obtained energy. Here, in our computational cell of 100 nm × 100 nm × 100 nm three NPs receive the same fluence, and after 200 ps we obtained the fragmentation of 10 nm NP only. So, we take a close look at its evolution in order to investigate the fragmentation mechanism. Namely, we must establish the connection of atomistic representation of the system with its microscopic parameters evolution (temperature, pressure, and density). In Figure 1a we show the atomic snapshots for selected times of 2 nm thickness slice of our MD system across 10 nm NPs, where water atoms indicated by blue color and the Si atoms are in red. Such the way of presenting the system evolution in the form of slices across the investigated area allow us to perform a visual analysis of the fragmentation process evolution in direct observation of the Si atoms interaction with water molecules in XY plane. Moreover, the consideration of temperature and pressure interplay, as shown in Figure 1b,c, upon the laser pulse absorption allows us to isolate the conditions determining the future evolution of the NP. As a result of laser heating, by 3 ps, the NPs gain a temperature of 11,500 K (above the critical value 5000 K [54]). Since the heating process occurs in liquid media, its mechanical resistance prevents the relaxation of internal stresses in the NP, and the pressure ~7 GPa is developed as a result of heating under the constant volume conditions by 3 ps. The accumulated stresses eventually relax with propagation of a spherical shock wave outward and rarefaction wave inward the fragmenting NP, visible in Figure 1c by light blue color at 10 ps. This compressive/rarefaction waves interplay indicates the onset of the cavitation bubble formation. This bubble growth is accompanied by fragmentation of vaporizing NP and immediate nucleation of the vapor to liquid clusters at the interface of Si vapor and water. This fast nucleation is facilitated by significant thermal losses due to heat of vaporization of liquid at 30 ps. By that time, the cavitation bubble expansion is balanced by the increased pressure of surrounding liquid volume, the particles of water (in liquid and vapor phase) penetrate the bubble volume, increasing therefore, the cooling efficiency and facilitating nucleation of liquid clusters of Si, speeding-up the formation of new NPs. Then, the excessive pressure in liquid, surrounding the fragmenting NP, reverts the cavitation bubble expansion and we can see that by 100 ps it collapses with the corresponding jump in pressure, ~5 GPa, and temperature ~3500 K in the center. Meanwhile, the bubble collapse leaves the fragmented NPs behind. They are trapped by liquid media and quickly undergo to solidification process due to fast cooling. Eventually, by 250 ps we can observed several completely formed fragmented NPs of spherical shape and more or less similar size of 2–5 nm in diameter. For a better visual observation, in Figure 1a last framed snapshot at 250 ps, the obtained fragments are shown in the form of 2 nm slice across the same volume but in XZ plane with water particles blanked. It should be noted here that unlike the monolithic NP, the initially porous NP of the same 10 nm size did not fragment under the same conditions. The NPs of 20 nm and 30 nm size did not fragment in both cases. From here we can conclude that the applied fluence of 0.015 J/cm^2^ is close to the threshold value for the fragmentation of 10 nm NPs.

Next, in order to clarify the effect of porosity and the incident fluence on the mechanism of NPs fragmentation process, we perform two simulations of fs laser pulse interaction with 3 monolithic and 3 porous NPs of the same 10 nm size in water, but at three different fluences of 0.015 J/cm^2^, 0.06 J/cm^2^, and 0.135 J/cm^2^, Figure 2. Basically, we use the same computational cell as it was used in the previous simulation, but this time the located inside NPs receive different fluences within the same computational geometry from virtually three different pulses. Three virtual pulses of different energy are combined into the single computational setup from the point of efficiency of calculations. The porous NPs, however, are placed in a separate computational cell. The porous NPs contained from 1 to 3 randomly placed pores to model the porosity of ranging 15–30%. The results of both simulations are presented as a sequence of atomic snapshots of Si atoms at selected times: 10 ps, 30 ps, 50, 100 ps, 150 ps, and 250 ps. The atoms of Si are colored by their potential energy as it is a handy indicator of their local ambient (interaction with neighbors) and can be therefore used as a phase indicator(red—gas, light blue—liquid, dark blue—solid). The water particles are not shown for a better visual analysis. It can be reconfirmed here that the fluence of 0.015 J/cm^2^ can be referred to as a threshold value for 10 nm NP fragmentation.

According to our model, the absorption cross section of the NP is equal to its geometrical value. As a result, having the same diameter, the porous NP gets a bit more energy than a monolithic one since the same amount of absorbed energy is distributed on fewer atoms. However, while analyzing the porous and monolithic 10 nm NP evolution from Figure 2, it seems that the initially higher kinetic energy of the expanding atoms (due to ablation via the phase explosion mechanism) is then compensated by less momentum of the expanding cavitation bubble against the same resistance of the surrounding liquid. Moreover, we can see that in the case of porous NPs, for the same energy input, the liquid media easier suppress the expanding debris, and the cavitation bubble collapse occurs faster, preventing liquid or vapor from penetrating inside the bubble volume. Subsequently, the faster collapse results in merging fragments of the NP back into a whole piece (neglecting the vaporized mass losses left in the surrounding liquid as single atoms or droplets much smaller than 1 nm). It can also be seen from Figure 2b at 50 ps, that due to the influence of the shock waves coming from the other fragmenting NPs, the merged NP gains momentum and moves across the computational cell volume.

A more beneficial final picture can be observed for the NPs fragmenting at higher energies of 0.06 J/cm^2^ and 0.135 J/cm^2^. Namely, it is clearly visible that the resulting fragmented NPs’ average size is a strong function of the absorbed energy. The above discussions about the general mechanism of an NP fragmentation process apparently remain valid for higher fluences. The original NPs are heated during the electron-phonon equilibration process, which is assumed to be 1 ps in these simulations. The strong heating results in a temperature significantly above the critical point (more than 10,000 K), reached under the conditions of constant volume due to significant resistance from the side of the liquid. As a result, the maximum internal pressure values are 14 GP and 25 GP for the fluences of 0.06 J/cm^2^ and 0.135 J/cm^2^, correspondingly. The subsequent fragmentation process, therefore, occurs due to the ablation via the explosive boiling mechanism and the process of expansion-contraction of the cavitation bubble. Then, the cooling and nucleation of new fragmented NPs occurs due to high thermal losses during the massive vaporization of the surrounding liquid.

Therefore, the resulting difference (fragmented and not fragmented) is obtained for the threshold fluence only, 0.015 J/cm^2^. For higher fluences, we observe that the obtained fragments between both monolithic and porous NPs are very similar in size for each of the applied fluences. Although we did not recognize any noticeable difference between the fragments of initially monolithic and porous NPs, this finding can potentially give us a reliable tool in succeeding in obtaining the NP size and size distribution of the required values demanded in biomedical applications [55,56].

For a more quantitative analysis of the fragmented NPs, we are planning several more large-scale simulations in the future. Since in general the fragmentation process even under the same irradiation conditions can have an arbitrary result, the characterization of the fragmented NPs by their size distribution can give us a more rigorous tool for the development of the methodology for laser-assisted generation of NPs with required properties. However, in this case, the initial computational cell must be modified. In the presented modellings we used high concentration of the initial colloidal solution, 3.5 mg/mL, which is however in the range of concentrations used in biomedical applications 0.1–10 mg/mL [57,58,59,60]. This assumes, and it was confirmed in our simulations, that the process of fragmentation is influenced by the interfering shock waves propagating from neighboring fragmenting NPs. To isolate the fragmenting processes between neighboring NPs, a more suitable computational cell for quantitative analysis of the obtained fragments must include non-reflective boundaries (NRB). These are dynamically behaving boundaries devoted to absorbing the incoming pressure waves without any reflection (the use of ridged or free boundaries results in the reflected waves anyway). The construction of NRB boundaries was suggested in [61], and they were designed for the absorption of planar pressure waves. While this type of NRB was frequently utilized in a number of nanostructuring simulations [62,63], the ablation mechanism during the NP fragmentation process discussed in this work assumes the development of spherical pressure waves (shock waves due to cavitation bubble formation). In this case, a more accurate Langevin-NRB, aimed at absorbing non-planar laser-induced pressure waves, has been recently suggested in Ref. [64]. Alternatively, the NRB boundaries of spherical shape can be constructed in a similar way, as they were of cylindrical geometry, as suggested in Ref. [65].

Finally, in order to model the fragmentation process of the colloidal solutions of realistic concentration, the series of simulations of the laser pulse interaction with a single NP in the center of a spherical computational cell of 100 nm in diameter and larger must be performed. With the total computational cell modified accordingly and the laser light absorption process described via the Maxwellian system [63,66], the corresponding large-scale simulations of ultrashort laser pulse fragmentation of NPs of different initial size and different porosity and at different pulse durations and the incident fluences can result in the development of the methodology for laser-assisted formation of NPs with predesigned properties.

## 3. Materials and Methods

### The Model and Computational Setup

In order to investigate the mechanism of the laser-induced fragmentation process of Si NPs, we performed large scale parallel simulations of fs laser pulse interaction at the incident fluence of 0.015 J/cm^2^. The irradiated volume is represented by cube of water 100 nm × 100 nm × 100 nm, where 3 NPs of 30 nm, 20 nm, and 10 nm are submerged. The scheme of the corresponding computational cell is shown in Figure 3. The entire computational cell can be processed in multiprocessor mode with division into N_x_, N_y_, and N_z_ processors (in our case 8 × 8 × 6 = 384 processor cores) in X, Y, and Z directions, respectively. The total energies of the subsystems of silicon E_Si_ and water E_H2O_ were described by the Schillinger-Weber potential (SW) for silicon [67] and the embedded atom method (EAM) for water [68], which mainly reproduces its mechanical properties. The energy of Si-water interactions E_Si-H2O_ at the interfaces of two media is given via the Lennard-Jones potential, parameterized based on the values of the Van der Waals radius of silicon and oxygen atoms (as the largest in a water molecule) [69] and the fact that silicon is not wettable for estimation of adhesion-related parameters. As an assumption, the latter results in an adhesion coefficient as small as four times compared to that of water-water interaction.

Two computational cells containing Si matrix and water were equilibrated at 273 K. Then, three NPs of 10 nm, 20 nm, and 30 nm were cut off the Si matrix and embedded into the water media with removal of those atoms that have positive potential energy along the Si-water interface. The complete system was then equilibrated once again at the initial temperature 273 K. All three NP were placed at roughly equal distances between each other, and the periodic boundary conditions were applied in all three directions, imitating the colloidal solution of high concentration (~3.5 mg/mL). The porous NPs were prepared by means of cutting a few spherical pores (1–3) of random size and random position inside the NPs.

In order to describe the laser pulse interaction with colloidal NPs in our simulations we assumed that the absorption cross section of the irradiated NPs is equal to its geometrical cross section and NPs absorbs all the incident energy homogeneously. According to the Mie theory [70,71], this means that the main contribution to absorption is provided by the zero harmonic, and the absorption of light dominates over the elastic losses in the interaction of light with matter. Such an approach can be a useful tool for observing the kinetics of molecular dynamic systems at the conditionally upper limit of the light absorption efficiency, for example, when it is assumed that all incident radiation is converted into the internal energy of NPs.

Here it is worth mentioning that the linking our results to a particular wavelength is conditional. In fact, the solution of the wave equation is not implemented in the modeling. Our work is basically devoted to the analysis of fragmentation mechanism, so we took the absorption efficiency equal to 1 (according to the Mie Theory). That is, the fragmentation thresholds found in our simulation must be increased in proportion to the absorption efficiency for a particular wavelength. For instance, at the wavelength of 340 nm, this factor is 20.

It is also assumed here that upon the laser light absorption due to laser-assisted generation of free carriers, the lattice is heated within 1–3 ps time (the characteristic time of the electron-phonon energy equilibration in Si [51]). The process of absorption, therefore, was modelled by the Berenson thermostat method [72], where the laser energy distributed among the atoms constituting a NP was delivered via scaling of the velocities in the correspondence with the Gaussian profile of a 1 ps laser pulse. It should be noted here that in real systems, part of the energy is always scattered and reflected when interacting with NPs. Moreover, if the resonant absorption occurs, the absorption efficiency can be even greater than 1. A more accurate solution of the NP’s absorption process can be given by Maxwell equations [73], considering the multipole absorption of light by nanoparticles depending on the wavelength.

In addition, we do not account for possible ionization processes, and the fragmentation of NPs due to a Coulomb explosion is not available in our simulation [45,46]. Thus, in our modeling, we investigate solely the effects of the water environment, which are mechanical resistance and thermophysical properties. This investigation, however, is an important step forward towards understanding the physical processes underlying the laser-assisted fragmentation of NPs and the possibility of obtaining a colloidal solution of Si NPs with predesigned properties. Finally, the simulation was run across 384 processor cores, and the evolution of the entire computational cell is described in 3D with a resolution of macroscopic parameters (temperature, pressure, and density) of ~1 nm^3^.

## 4. Conclusions

In this first study of ultrashort laser pulse fragmentation of Si NPs, we reported our preliminary results obtained with the help of an MD-based model, which explicitly represents the interaction of Si atoms and water particles. Using the simple approximation of the laser light absorption process by NPs and the EAM potential for a description of the interaction between the water particles, we investigated a general mechanism of the colloidal NP fragmentation process. We found that strong heating under the conditions of water confinement (under constant volume) results in the onset of the ablation process via a phase explosion mechanism, which decomposes the irradiated NPs into vapor atoms with simultaneous expansion of the cavitation babble. Then, the relaxation of the laser-induced stresses via the shock wave propagation is balanced by the external pressure from the liquid, which ceases the Si vapor-waterfront interface, and the water particles start penetrating inside the cavitation bubble. Finally, due to significant thermal losses of Si atoms to the heat of vaporization of water particles, the nucleation of vapor Si atoms to clusters of Si material and their solidification completes the process of fragmentation.

The important finding from our simulation is the possibility of laser-induced fragmentation of NPs under the controlled conditions. Up to now, we have found that the average size of the final fragments is a function of the applied fluence, which is clearly seen from our simulations. With accounting based the approximations of laser-light absorption and the MD water representation, however, the direct comparison of the modeling results with the experimental data is hampered at this stage. Thus, the computational approach used in this work aims for identification of the general effects involved in the fragmentation process, solely caused by the mechanical action of the liquid media (water). Such a general approach, however, is an important intermediate step, and we can clearly isolate the physical mechanisms governing the initial NP fragmentation process and formation of new colloidal fragments. The appropriately modified computational cell along with an accurate description of the laser light absorption will bring the suggested model to a forefront level. The completely elaborated methodology in this case will give us a powerful tool for generation of colloidal solutions with predesigned properties in industrial and bio-medical applications.

## Figures and Tables

**Figure 1 ijms-24-14461-f001:**
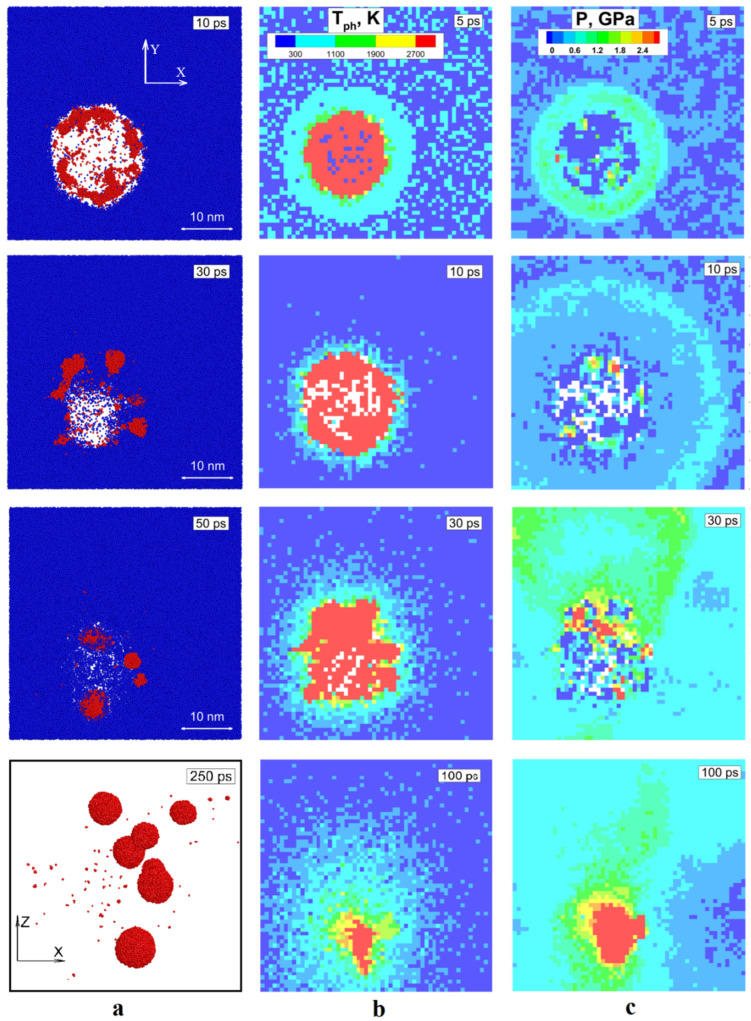
Evolution of the irradiated NP is shown as a sequence of atomic snapshots of 2 nm layer across the volume containing 10 nm NPs. The particles of water are colored by blue and Si atoms are in red. The last snapshot for a better visual is shown in XZ plane, whereas the rest are in XY. The water particle are blanked in the last framed snapshot at 250 ps. (**a**); The evolution of temperature field is shown by color for selected times for the same volume (**b**); the evolution of pressure is shown by color for selected times (**c**).

**Figure 2 ijms-24-14461-f002:**
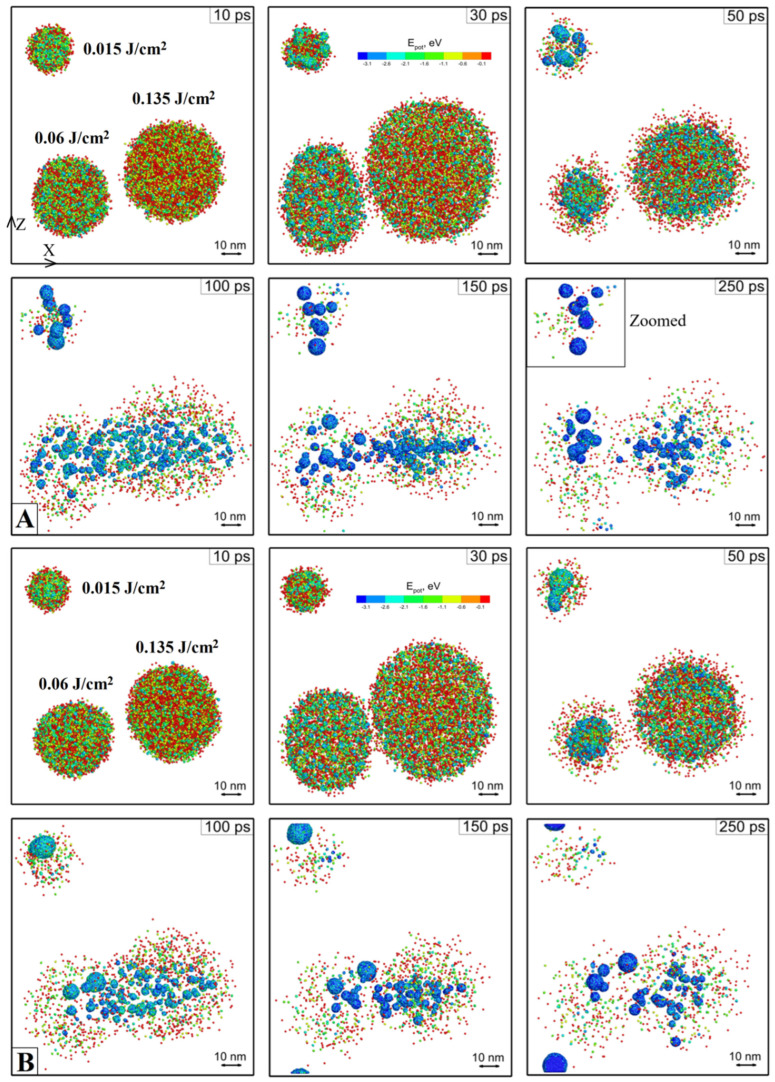
The atomic snapshots of Si NPs are shown for selected times for the case of monolithic (**A**) and porous (**B**) NPs. The Si atoms are colored by their potential energy for a handy identification of gaseous and liquid phases. The water particles are blanked for a better visual analysis. The squared area in (**A**) at 250 ps is zoomed and analyzed in Figure 1a. atomic snapshots of Si NPs are shown for selected times for the case of monolithic (**A**) and porous (**B**) NPs. The Si atoms are colored by their potential energy for a handy identification of gaseous and liquid phases. The water particles are blanked for a better visual analysis.

**Figure 3 ijms-24-14461-f003:**
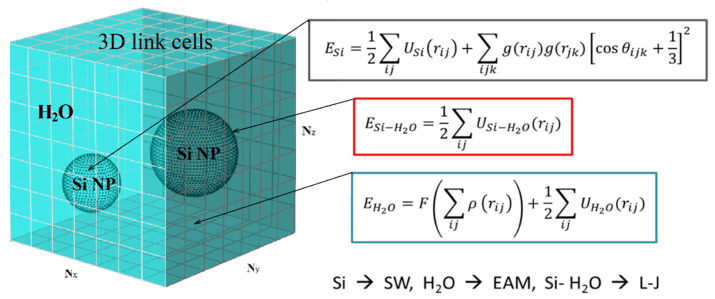
Schematic representation of the computational cell for modeling the process of laser disintegration of silicon NPs in a colloidal solution.

## Data Availability

The performed simulation results and obtained data is available by request.

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
