# Peer review of "Molecular Dynamics Modeling of Pulsed Laser Fragmentation of Solid and Porous Si Nanoparticles in Liquid Media"

_ijms, 2023, doi:10.3390/ijms241914461_

Round 1

Reviewer 1 Report

The paper “Molecular Dynamics Modeling of Pulsed Laser Fragmentation” written by Kutlubulatova et al

The reviewer reports

Comments

Line 128.  the atomic snapshots 128 for selected times of 2nm thickness slice of our MD system across 10 nm NPs” it would be good if the authors would explain this phrase with the help of a figure. It is important for understanding the relation of Figures 2 and 3 to the real three-dimensional flow.

Line 133. “As a result of laser heating, by 3ps the NP gain the temperature of 11500 K” Explain how the nanoparticle (NP) is heated. Otherwise the indication of the time moment of 3 ps is meaningless. Explain exactly in this place.

I approve the structure of the paper when first the results are given (Chapter 2) and then the technical details (Chapter 3).

But at this point you should say that

(1) the energy absorbed is equal to the incident fluence J/cm2 multiplied by the cross section of the particle.

(2) that the release of this energy is homogeneous over the substance of the particle.

(3) that the energy release is connected with electron-phonon coupling and lasts from the 1st to the 3rd picosecond.

That is, the actual beginning of heating and expansion of the particle takes place at time 1 ps.

Line 136. “The accumulated stresses eventually relax with propagation of a spherical shock wave,” I don't agree with that statement. It is a question of stress inside the particle. Relief/unload of this stress occurs in the rarefaction wave running from the surface of a spherical particle to its center.

Line 143 “and along with nucleation of Si  vapor into liquid clusters the particles of water (in liquid and vapor phase) penetrate the bubble volume, increasing therefore, the cooling efficiency and speeding-up the formation of new NPs.

There is both silicon liquid and silicon vapor together with liquid (water) and water vapor. Please write more clearly what is meant. Where is solid-liquid-gaseous silicon and where is liquid water and water vapor.

Otherwise, the word liquid is used for both molten silicon and water. Confusion arises

Fig. 2 (c) t=100 ps.

It's unclear what the high-pressure stain/spot (red spot) is associated with at such a late date. Perhaps/apparently a color palette error? Pressure should be low at late times.

Line 163 “two simulations of 270 fs laser pulse” Why do the authors confuse the reader by specifying a pulse duration of 270 fs? In reality, the heating and expansion of silicon starts from the 1st picosecond and lasts for about 2 ps. The processes related to the formation of electron-hole plasma + associated with dynamic effects from electron pressure and from the energy transfer of the electron subsystem into the lattice of the silicon crystal are not taken into account in the numerical code.

Line 168 “The atoms of Si are colored by their potential energy as it is a handy indicator of their local phase (red – gas, blue – liquid, green - surface).” It is not clear what the word "surface" means? It is a geometric cross-section of the particle volume, i.e. atoms in the volume are shown. Maybe the authors wanted to say solid state (solid) in this place?

Line 188 “due to the influence of the shock waves coming from the other fragmenting NPs the merged NP gains a momentum and moves across the computational cell volume.

Here it is necessary to explain from where neighboring particles appear. Explain that it is due to periodic boundary conditions in the computational box 100*100*100 nm3.

That is, the distance between neighboring particles in a periodic lattice of particles is 100 nm. A weak shock in water travels at a speed of 1.5 km/s = 1.5 nm/ps. That is, the wave from the neighbors should arrive after a time interval of 100 nm/1.5 nm/ps = 70 ps. But at the initial stage the shock wave is stronger and travels faster than the speed of sound.

Line 228 “high concentration of the initial colloidal solution, 3.5 mg/ml

taking the mass of the particle and relating it to the mass of water in a cube (100 nm)**3, we get concentrations of 1.2, 9.6, and 32.5 mg/mL for particles with diameters of 10, 20, and 30 nm, respectively. Where did the figure of 3.5 mg/mL come from?

Line 255 “processes of heat conduction leads” how is the thermal conductivity of water taken into account?

Line 268 “with division into Nx, Ny, and Nz processors in 268 X, Y, and Z directions

Then the number of processors is equal to the sum of Nx+Ny+Nz. Actually it seems that the numbering of processors is Nx,y,z, where "x,y,z" are indices in corresponding directions. Then the number of processors is equal to the product of the maximum values of these indices.

Line 291 “we assumed that the absorption cross section of the irradiated NPs is equal to its geometrical cross section” According to Mie theory, to the best of my knowledge, the absorption cross section of radiation (for particles smaller than the wavelength of EM radiation) is proportional to the linear size of the particle.

Line 298 “We can also relate our modeling to the incident wavelength of 800nm, where the absorption efficiency is equal to 1.0 for the NP of 20nm diameter, which corresponds the incident fluence of 0.02 J/cm2.Explain what this phrase means?

Comments/misprints

-affiliations are written by different styles

-in many places it is written “10nm” together and even at the same place the “10 nm” are separated. See, e.g., lines 124, 125 or 133-134

-numeration of Figures begins with Fig. 2. Where the Fig. 1 is?

-line 179 “and monocytic 10 nm NPs

-line 199  14 GP àGPa

-line 267 “cell is shown in Figure 1. 

Figure 1 is cited after Figures 1 and 2. Apparently, in the first version of the paper, Chapter 3 in the current version preceded Chapter 2 in the current version.

CONCLUSION

I believe the article requires considerable revision before publication

Reviewer 2 Report

In this study, authors conducted extensive Molecular Dynamics (MD) simulations to model the interaction between a femtosecond (fs) laser and a colloidal solution containing silicon nanoparticles (Si NPs). Subsequently, the generated nanoparticles were subjected to a detailed analysis of their shape and morphological attributes. The analysis enabled to draw conclusions regarding the varying contributions of distinct laser-induced processes and material properties to the mechanism underlying the generation of these nanoparticles.  This article can be categorized as an intriguing research study.

Some suggestions and questions for the author's consideration are as follows:

1. Please confirm the sequential numbering of figures in the main text: 2, 3,        1. Is that correct?

2. What are some other methods for generating nanoparticles besides Pulsed Laser Ablation in Liquids?

3. How do laser-synthesized silicon nanoparticles compare to other contrast agents for bio visualization?

4.  Can you explain the process of Pulsed Laser Ablation in Liquids and how it produces nanoparticles?

5. Figure 1,.Fig. 2, Fig. 3, the format is inconsistent.

no comment.

Round 2

Reviewer 1 Report

The article is sufficiently corrected and can be accepted in print as is